physiology

early life history, nutrition, bleaching, confocal microscopy, recruit, heterotrophy

# Feeding and thermal conditioning enhance coral temperature tolerance in juvenile *Pocillopora acuta*

Ariana S. Huffmyer[1,2], Colton J. Johnson[1,3], Ashleigh M. Epps[1,4], Judith D. Lemus[1,†] and Ruth D. Gates[1,†]

[1]Hawai'i Institute of Marine Biology, University of Hawai'i at Manoa, Kane'ohe, HI 96744, USA
[2]Department of Biological Sciences, University of Rhode Island, Kingston, RI 02881, USA
[3]Department of Biology, San Diego State University, San Diego, CA 92182, USA
[4]Department of Life Sciences, Texas A&M University-Corpus Christi, Corpus Christi, TX 78412, USA

ASH, 0000-0002-9170-0971; CJJ, 0000-0001-8895-6813; AME, 0000-0003-1873-0004; JDL, 0000-0001-6056-1262

Scleractinian corals form the foundation of coral reefs by acquiring autotrophic nutrition from photosynthetic endosymbionts (Symbiodiniaceae) and use feeding to obtain additional nutrition, especially when the symbiosis is compromised (i.e. bleaching). Juvenile corals are vulnerable to stress due to low energetic reserves and high demand for growth, which is compounded when additional stressors occur. Therefore, conditions that favour energy acquisition and storage may enhance survival under stressful conditions. To investigate the influence of feeding on thermal tolerance, we exposed *Pocillopora acuta* juveniles to temperature (ambient, 27.4°C versus cool, 25.9°C) and feeding treatments (fed versus unfed) for 30 days post-settlement and monitored growth and physiology, followed by tracking survival under thermal stress. Feeding increased growth and resulted in thicker tissues and elevated symbiont fluorescence. Under high-temperature stress (31–60 days post-settlement; *ca* 30.1°C), corals that were fed and previously exposed to cool temperature had 33% higher survival than other treatment groups. These corals demonstrated reduced symbiont fluorescence, which may have provided protective effects under thermal stress. These results highlight that the impacts of feeding on coral physiology and stress tolerance are dependent on temperature and as oceans continue to warm, early life stages may experience shifts in feeding strategies to survive.

[†]Joint senior authors.

**Author for correspondence:**
Ariana S. Huffmyer
e-mail: ashuffmyer@gmail.com

# 1. Introduction

Nutritional flexibility (i.e. mixotrophy) is a strategy that allows organisms to acquire carbon and other essential nutrients through multiple nutritional modes in order to survive in variable and stressful conditions [1]. In coral reef ecosystems, calcifying scleractinian corals use species-specific strategies including flexibility in autotrophic (photosynthesis) and heterotrophic (feeding) nutritional modes to survive in nutrient-limited tropical oceans [2–4]. The nutritional symbiosis between corals and photosynthetic endosymbionts (Symbiodiniaceae [5]) has facilitated the ecological success of reef-building corals by supporting greater than 90% of coral host energetic demand through photosynthetically fixed carbon [6,7]. In exchange, the symbiont assimilates metabolic waste (e.g. inorganic carbon, nitrogenous waste) from the host, resulting in tight nutrient recycling within the holobiont [8,9]. However, symbiont-derived nutrition is deficient in essential nutrients (e.g. nitrogen and phosphorus) [10] and coral feeding—ranging from the uptake of dissolved nutrients to polyp feeding on plankton—allows corals to obtain additional essential nutrients that are needed to support growth and calcification (reviewed in [4]). Access to adequate prey supply (i.e. plankton) has a strong influence on the physiology and performance of reef-building corals. Notably, fed corals exhibit increased tissue growth [11,12], protein concentrations [12,13], lipid stores [14,15] and calcification rates [16–18]. Feeding also benefits Symbiodiniaceae endosymbionts by increasing symbiont population density [16], photosynthetic rate [13] and efficiency [11,19] as well as photopigment concentration [12,13], which can improve the quality and quantity of translocated carbon to the coral [20].

Under stressful conditions, feeding can provide an important source of nutrition and can increase stress tolerance in adult corals. Ocean warming causes more frequent and severe coral bleaching events that compromise the coral-algal nutritional symbiosis through the loss of symbiotic algae and/or algal pigments [21,22]. In the face of recurrent stressors, such as bleaching, that compromise the coral's primary autotrophic symbiotic nutrition source, environmental and nutritional conditions that facilitate the acquisition of energy and biomass storage may support stress tolerance and survival [3,23]. Coral respiratory demand and the need for additional energy acquisition increases under thermal stress, including increased metabolic rates [24,25] and the activation of cellular stress responses [26–28]. During severe thermal bleaching, corals therefore enter a state of starvation unless feeding [7,19] and/or utilization of biomass reserves [29,30] compensates for the loss in symbiotic nutrition [23]. Previous work has found that supplemental energy acquired through feeding supports survival [7,23] and offsets the costs and reduces negative downstream consequences from cellular stress [31]. Corals that feed have a greater capacity to resist bleaching [32], accelerate recovery [33], promote recolonization of symbiont communities [34], maintain symbiont photosynthetic efficiency during bleaching [35] and re-establish nutritional exchange during recovery [36]. Feeding also provides an important source of essential metals to symbiont populations, which helps corals to resist bleaching under high temperatures [37].

Coral tissue thickness is a key trait associated with thermal tolerance, providing enhanced energy reserves and protection for symbiont communities during bleaching [7,38,39]. Therefore, exposure to conditions that facilitate tissue biosynthesis and storage, including feeding, could enhance bleaching tolerance by increasing available energy reserves for use under subsequent stress [39–41]. In addition to nutritional inputs, favourable thermal regimes can also facilitate biomass storage—at cooler temperatures, respiratory demand declines and may provide an opportunity for excess energy acquired through symbiosis and feeding to be stored in tissue [39,42,43]. For example, during natural cool seasonal cycles (i.e. winter), biomass, chlorophyll concentrations and Symbiodiniaceae densities peak, then decline during warmer months when host respiratory metabolism increases in response to elevated temperature [42,43] and exposure to cool temperature has been shown to affect coral response to thermal stress [39,44,45]. However, our understanding of interactive effects between temperature and feeding on coral biomass and performance is limited.

Most feeding studies in corals have focused on adult colonies, leaving the influence of feeding poorly understood in the juvenile stage. Feeding behaviour begins in coral early life history after metamorphosis and formation of polyp and tentacle structures that enable particle capture [46]. During this period of nutritional transition, juveniles reduce reliance on parentally provisioned energy resources and increase utilization of symbiont nutrition and heterotrophic feeding [47–50]. The survival of early life stages of corals is critically important for reef replenishment [51,52]; however, high mortality of early life stages is further exacerbated by environmental stress (e.g. ocean warming), presenting roadblocks for reef persistence as marine heat waves become more frequent and severe [53–55]. During

energetically vulnerable life stages (i.e. post-settlement juveniles), feeding may provide a strategy that supports growth and survival during recruitment under stress by providing supplemental energy and facilitating biomass storage. Although previous work has documented that feeding promotes growth and survival of post-settlement corals [56–58], there is a need to directly test the effects of feeding on thermal tolerance in early life stages and understand the role of temperature in shaping feeding effects.

Here, we hypothesize that feeding, combined with a period of preparation for stress through prior exposure to cool temperature, will facilitate biomass storage in juvenile corals and enhance survival when exposed to thermal stress. We exposed *Pocillopora acuta* juvenile colonies to two thermal regimes (ambient, 27.4°C; cool, 25.9°C) either in the presence or absence of freshly collected natural plankton for 30 days post-settlement. We applied the use of laser scanning confocal microscopy (LSCM) to characterize juvenile tissue thickness and Symbiodiniaceae fluorescence to track both host and symbiont responses to thermal and feeding treatments. We then tested the influence of these thermal and feeding treatments on juvenile temperature tolerance by tracking survival during a subsequent 30-day period of high-temperature stress (*ca* 30.1°C). We found that fed *P. acuta* juveniles exposed to cool temperature prior to a thermal stress had the highest survival under high temperature.

# 2. Material and methods

## 2.1. Larval collection

Coral juveniles used in this experiment were settled from larvae collected from adult *Pocillopora acuta* (Lamarck, 1816; electronic supplementary material, figure S1A). Adult colonies ($N = 24$, 15–20 cm diameter) were collected from the reef crest (lat. 21°26′03″ N, long. 157°47′12″ W; 1–3 m depth) at the Hawai'i Institute of Marine Biology (HIMB, patch reef #1), O'ahu, Hawai'i, under the HIMB Special Activities Permit 2018–03 (Division of Aquatic Resources, Hawai'i). Planula larvae were collected from adult colonies in the full moon period over four nights of planulation in June 2017, during the summer season (electronic supplementary material, figure S1B). Larvae were pooled prior to settlement with larvae released during the same night identified as a 'cohort'. Larvae were then settled on acrylic plugs in treatment tanks (described below; electronic supplementary material, figure S1C). Plugs were conditioned for five months in outdoor tanks supplied with unfiltered raw seawater under ambient light prior to settlement to allow for the establishment of crustose coralline algae. Each day, pooled larvae were allocated into mesh-top settlement containers submerged in treatment tanks either in cool or ambient temperature treatments (electronic supplementary material, figure S2; $n = 20$ plugs per container, $n = 2$ containers per tank, $n = 6$ tanks per temperature treatment). After 24 h, plugs with settled juveniles were moved to racks in the bottom of tanks (electronic supplementary material, figure S1D).

## 2.2. Rearing treatment conditions

Coral juveniles were exposed to temperature and feeding treatments for 30 days post-settlement (hereafter referred to as 'rearing period'). Rearing period temperature treatments began at larval settlement with feeding treatments starting one day following settlement. During settlement, we monitored the settlement rate on a subset of plugs in controlled trials to examine whether temperature treatments (described below) influenced settlement. Fifteen larvae were placed in an isolated settlement container with one settlement plug ($n = 15$ larvae per trial, $n = 3$ trials per tank). Settlement rate was calculated as a proportion of larvae settled after 24 h.

Once larvae had settled, juveniles were exposed to temperature (cool versus ambient) and feeding treatments (fed versus unfed) for 30 days followed by a subsequent 30-day simulated high-temperature stress (electronic supplementary material, figure S2). Experimental tanks ($N = 12$, 40 l) were supplied with flow-through filtered (1 µm) chilled seawater at 290–360 ml min$^{-1}$ (replacement rate 1.9–2.3 h) equipped with a recirculating pump (Rio 1100+ at maximum rate) and aquarium lighting (LED xr30w Pro, EcoTech Marine, Allentown, PA, USA) programmed with diurnal ramping on a 12 L : 12 D h cycle, peaking at approximately 235 µmol photons m$^{-2}$ s$^{-1}$ (LI-192 quantum sensor, LI-COR Biosciences, Lincoln, NE, USA).

Rearing temperature treatments were controlled using a Neptune Apex Aquarium Controller system in combination with a titanium submersible heater (TH Series 300 W, Finnex, Chicago, IL, USA) calibrated to a digital thermometer (Digi-Sense Thermometer, Traceable Products, Webster, TX, USA).

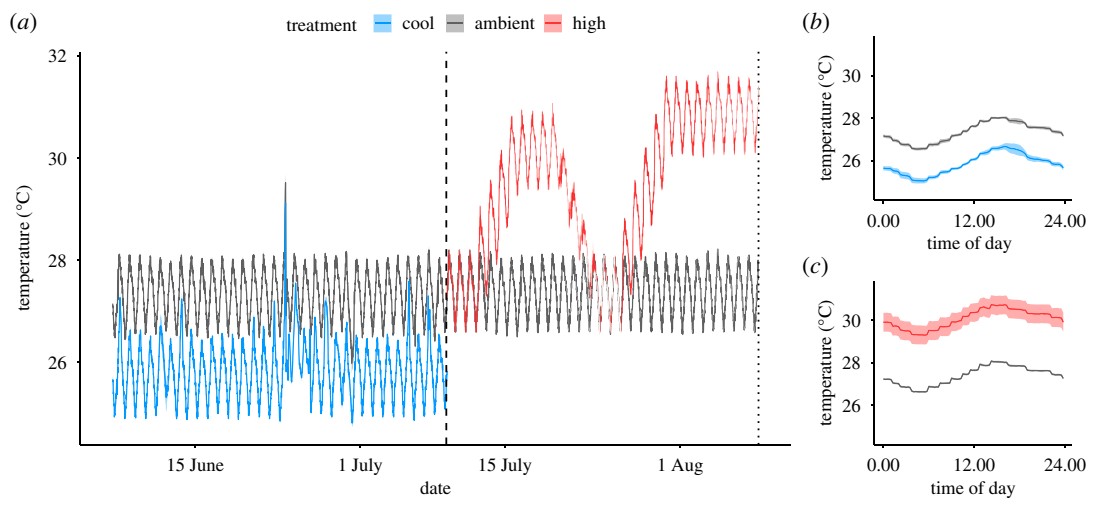

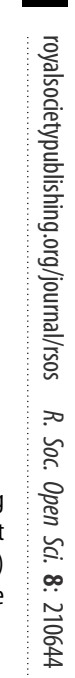

**Figure 1.** Temperature treatment profiles. (*a*) Mean temperature (°C) (blue = cool; grey = ambient; red = high) profile over rearing treatment period and thermal stress period in 12 tanks (*n* = 6 per treatment). Dashed line indicates the end of rearing treatment period and the start of thermal stress period. Dotted line indicates the end of the thermal stress period. (*b*) Mean temperatures (°C) recorded every 15 min over 24 h cycle during rearing period. (*c*) Mean temperatures (°C) recorded every 15 min over 24 h cycle during thermal stress period. In all plots, shading indicates standard error of mean.

The ambient and cool temperature treatments were based on the average temperatures in June 2016 and March 2016 in Kāne'ohe Bay, Hawai'i, respectively (Pacific Islands Ocean Observing System, Station 51 207; electronic supplementary material, table S1). Mean (±s.e.m.) temperature in the ambient treatment was 27.3 ± 0.2°C and 25.9 ± 0.3°C in the cool treatment (figure 1; electronic supplementary material, table S1). During the rearing period, the mean temperature in the cool and ambient treatments was within 0.4°C of temperatures in Kāne'ohe in March 2017 and June 2017, respectively (electronic supplementary material, table S1). The mean daily temperature range was higher in the experimental tank system (1.5–1.6°C in cool and ambient tanks compared to approximately 0.6°C in Kāne'ohe Bay; figure 1; electronic supplementary material, table S1). There was a rapid and short increase in temperature at rearing day 18, which was a 1 h failure of the temperature regulation system in which temperature in all tanks reached approximately 29.8°C (figure 1). There was no increase in juvenile mortality following this event.

Corals were provided with natural plankton assemblages three times per week during daylight (between 13 : 00 and 16 : 00 h) beginning 1 day after settlement (electronic supplementary material, figure S2). Size-fractioned plankton (63–243 µm fraction containing copepods, zoea and phytoplankton) were added to treatment tanks at a concentration of 2811 ± 77 plankters l$^{-1}$ (mean ± s.e.m.). Plankton were collected with a plankton net towed at 1–3 m depth at the same patch reef where parent coral collection took place in Kāne'ohe Bay (HIMB, patch reef #1; lat. 21°25′55″ N, long. 157°47′36″ W) [59,60] and were filtered and fractioned to isolate the 63–243 µm size class and remove the raw seawater and prey items outside the desired size class. Plankton were then rinsed with 1 µm filtered seawater prior to addition in tanks to ensure that corals were supplied with plankton during feeding without the addition of other dissolved or particulate nutrients from raw seawater. During feeding, aquarium lighting and water flow were turned off for 1 h in all fed and unfed tanks while maintaining tank circulation. In this study, we did not measure feeding rate or plankton ingestion rates and therefore we investigated the effect of access to prey on juvenile responses. Turf algal growth on tanks and on plugs surrounding juvenile corals was cleaned weekly. Rearing treatment groups in this experiment were ambient-unfed, ambient-fed, cool-unfed and cool-fed (electronic supplementary material, figure S2; *n* = 3 tanks per treatment).

## 2.3. Rearing period responses

During the 30-day rearing treatment period, survivorship was measured weekly as a binary response of the presence or absence of living juveniles on each plug (*n* = 165–192 plugs per treatment). Fifteen randomly selected plugs (*n* = 2–25 juveniles per plug) from each tank were photographed at the

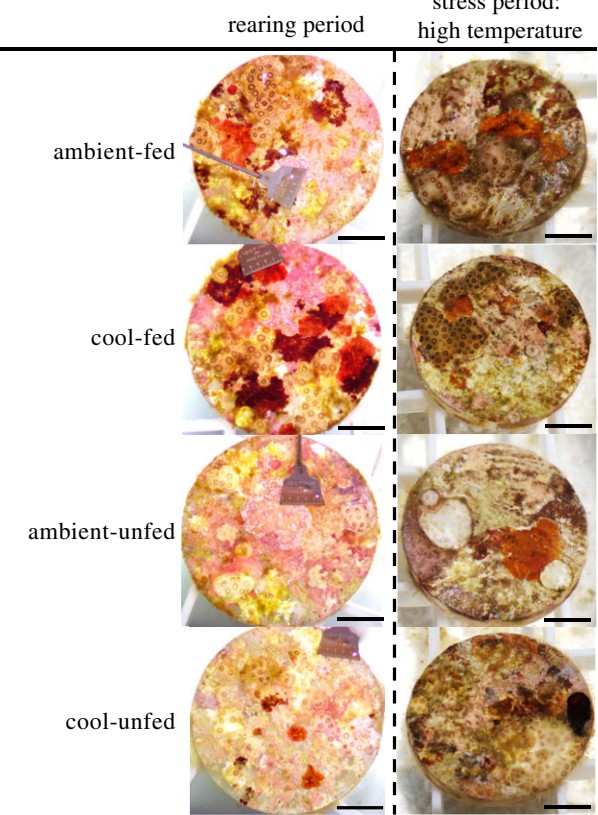

**Figure 2.** Photographs of juvenile corals at the end of the rearing period (left) and at the end of the stress period under high temperature (right). Rearing treatment groups listed vertically in order: ambient-fed, cool-fed, ambient-unfed, cool-unfed. Scale bars indicate 5 mm.

beginning and end of the rearing period using a stereomicroscope (Amscope, Irvine, CA, USA) for analysis of growth. Using Fiji/ImageJ software (v. 2.0.0), growth was calculated as the per cent change in planar surface area per day (% planar extension day$^{-1}$) of each colony ($n = 117$–$154$ per treatment) from the start to end of the 30-day rearing period. Because larvae in this experiment settled either as individuals or in fused aggregations, juveniles resulting from fused aggregations were removed from analysis of growth. Photographs of juveniles at the end of the rearing period are displayed in figure 2.

At the end of the rearing period, tissue thickness and Symbiodiniaceae fluorescence were measured on juvenile colonies from five randomly selected plugs per tank ($n = 22$–$36$ total juveniles per treatment). Tissue thickness and Symbiodiniaceae fluorescence were measured using LSCM with a Zeiss LSCM 710 confocal microscope and Zen Black (2011 v. 14.0.16.201) software with methods and settings as described previously for *P. acuta* [61]. Juveniles on plugs were fixed in 1 : 4 zinc-buffered formalin (Z-Fix Concentrate, Anatech, Ltd., Battle Creek, MI, USA) diluted with filtered seawater (1 µm) and refrigerated (4°C) to allow for processing at a later date. Briefly, autofluorescence of both the coral host and Symbiodiniaceae was used to characterize tissue thickness and Symbiodiniaceae fluorescence, respectively. First, the average depth of tissue fluorescence (µm) was measured on each colony as a proxy of tissue thickness [61] on three cross sections ($n = 3$ measurements per cross section) of three-dimensional rendered models. Second, the average intensity of fluorescence emitted by Symbiodiniaceae was measured on each colony ($n = 3$ measurements per colony), and these values were calibrated to relative intensity (RI; Red InSpeck Microscope Image Intensity Calibration Kit, Molecular Probes, Sigma-Aldrich, St. Louis, MO, USA). Previous work has shown that Symbiodiniaceae fluorescence measured by LSCM is positively related to both Symbiodiniaceae chlorophyll *a* content and population cell densities [61], and therefore, in this study, we use fluorescence as a proxy measure for these characteristics, which are related to the potential for energy acquisition and translocation by the symbiont population [62].

To further validate the use of LSCM methodology to measure tissue thickness and Symbiodiniaceae fluorescence in juvenile *P. acuta* corals, we conducted paired measurements of these responses with comparative methods—measurement of tissue thickness on tissue tunics (i.e. tissue of juveniles

following skeletal decalcification) and Symbiodiniaceae cell density using a hemocytometer, respectively (electronic supplementary material, figure S3), using methods modified from [61]. We chose to validate this methodology using measurements of fed and unfed colonies as previous research has demonstrated the strong effect of feeding on coral physiology [4] that allowed us to ground-truth LSCM measurements. Using one-way analysis of variance tests (ANOVA), we confirmed that tissue thickness ($p < 0.001$) and Symbiodiniaceae fluorescence ($p < 0.001$) measured by LSCM as well as tissue thickness ($p = 0.002$) and Symbiodiniaceae cell densities ($p = 0.036$) measured by comparative methods were enhanced in fed corals (electronic supplementary material, figure S3). In addition, paired measurements for tissue thickness ($p < 0.001$) and Symbiodiniaceae metrics ($p = 0.046$) were significantly positively correlated (electronic supplementary material, figure S3). These comparisons validate the use of LSCM to characterize juvenile coral tissue thickness and Symbiodiniaceae fluorescence in this study. Detailed methods and outcomes of these comparisons are available in the electronic supplementary materials.

## 2.4. Thermal stress period

At the end of the rearing period, juvenile thermal tolerance was measured by tracking survivorship during a 30-day thermal stress period. We exposed juveniles from each rearing temperature treatment to either ambient or high-temperature conditions and continued respective feeding treatments with methods as described above (electronic supplementary material, figure S2). Juveniles were assigned to the same feeding treatment during both the rearing and stress periods of this experiment. The ambient temperature treatment was a continuation of the same temperature profile from the rearing ambient treatment and was controlled as described above. In the high-temperature treatment, temperature was increased at a rate of 0.7°C per day until reaching the target temperature. Mean (±s.e.m.) temperature in the ambient treatment was $27.4 \pm 0.1$°C and $30.1 \pm 1.7$°C in the high treatment (figure 1a,c; electronic supplementary material, table S1). During the stress period, mean daily temperature range was similar to the rearing period with a mean value of 1.4°C for both ambient and high treatments (figure 1c; electronic supplementary material, table S1). Due to failures in the experimental temperature control system, the high-temperature treatment was interrupted after 8 days of exposure, at which time temperatures were returned to the ambient treatment (4 days of decreasing temperature and 2 days at ambient temperature). Temperatures were again increased to a second phase of high temperature over 4 days (0.7°C increase per day) and held for 10 days (figure 1a). While an interruption in the high-temperature treatment was not intended, the thermal stress profile of a dual-pulse stress reflects conditions that would occur on the natural reef due to storms or surface mixing [63].

Survivorship was measured by counting the number of living juvenile colonies on each plug at the beginning and end of the stress period ($N = 327$ plugs). Photographs of juveniles in the high-temperature treatment at the end of the stress period are displayed in figure 2.

## 2.5. Data analysis

All statistical analyses were conducted in R (v. 3.6.1) [64] for the following response variables: (i) larval settlement; (ii) post-settlement survivorship; (iii) post-settlement growth; (iv) tissue thickness; (v) Symbiodiniaceae fluorescence and (vi) thermal tolerance. Linear mixed-effect model analyses were conducted using the *lme4* package [65] unless otherwise specified. Larval settlement was analysed using a logistic mixed-effect model with temperature as a fixed effect and cohort and tank as random intercepts. Post-settlement survivorship was analysed using a Cox Proportional Hazards model with temperature and feeding as fixed effects in the *survival* package [66]. Juvenile growth was quarter root transformed to meet normality assumptions and analysed with feeding and temperature as fixed effects and cohort and plug nested within the tank as random intercepts. Tissue thickness and Symbiodiniaceae fluorescence were analysed with temperature and feeding as fixed effects with cohort and plug nested within the tank as random intercepts. We used linear regressions and Akaike information criterion scores (AIC) model selection to determine whether colony size should be included as a random covariate in the analysis of tissue thickness and Symbiodiniaceae fluorescence. Because there was a significant relationship between tissue thickness and colony size ($p < 0.001$) and the model fit was improved (as indicated by lower AIC scores), colony size was included in the analysis of tissue thickness as a random covariate. However, colony size did not have a significant relationship with Symbiodiniaceae fluorescence ($p = 0.271$) and did not improve model fit, and

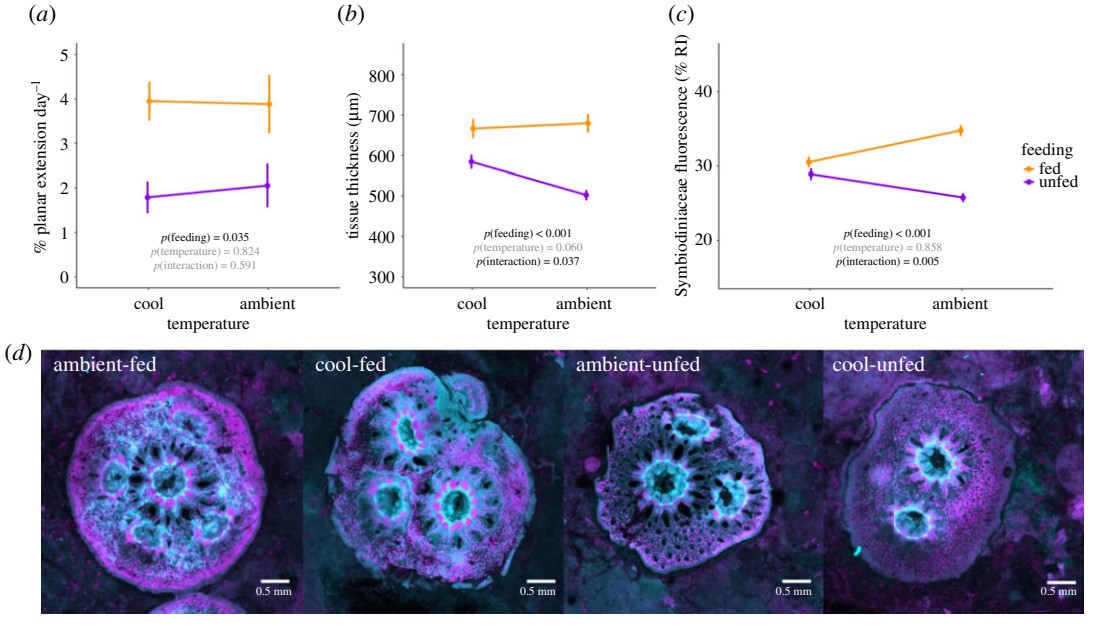

**Figure 3.** Juvenile coral responses to thermal and nutritional treatments. (*a*) Growth (% planar extension day$^{-1}$); (*b*) tissue thickness (µm); (*c*) Symbiodiniaceae fluorescence (RI) of juvenile *Pocillopora acuta* corals in rearing temperature (x-axis) and nutrition (purple = unfed; orange = fed) treatments. Error bars indicate the standard error of mean. *p*-values indicate significance (black text = $p < 0.05$, grey text = $p > 0.05$). (*d*) Representative examples of juvenile corals from each treatment under LSCM. Cyan indicates coral tissue fluorescence, magenta indicates Symbiodiniaceae fluorescence. Scale bars indicate 0.5 mm.

therefore, it was not included in the analysis. The relationship between tissue thickness and Symbiodiniaceae fluorescence was examined using a Pearson correlation.

Survivorship during the stress period was analysed with a logistic mixed-effect model with stress temperature, rearing temperature and feeding as fixed effects and cohort and tank as random intercepts. Normality assumptions were assessed using quantile–quantile plots and the homogeneity of variance was examined using Levene's tests in the *car* package [67]. Overdispersion in logistic models was assessed using the *blmeco* package [68]. Significance of fixed effects and their interactions were generated using Type II sum of squares ANOVA analyses in the *lmerTest* package [69]. Data and scripts to reproduce analyses and figures are publicly available [70].

# 3. Results

## 3.1. Rearing period

Larval settlement success was not influenced by temperature treatment ($\chi^2 = 0.32$, d.f. = 1, $p = 0.569$). Mean (±s.e.m.) settlement after 24 h was $42.3 \pm 3.1\%$ in ambient temperature and $39.4 \pm 3.2\%$ in cool temperature. Juvenile survivorship during the rearing period was 7% higher in cool temperature as compared to ambient temperature ($p < 0.001$; electronic supplementary material, figure S4 and table S2 and S3). Survivorship of juveniles during the rearing period was not influenced by feeding ($p = 0.473$; electronic supplementary material, figure S4) or an interaction between feeding and temperature ($p = 0.086$; electronic supplementary material, table S3). Juvenile growth rates increased from 1.9% per day in unfed juveniles to 3.9% per day in fed juveniles ($p = 0.035$; figure 3*a*; electronic supplementary material, table S2 and S3). There was no effect of temperature ($p = 0.824$) or the interaction between temperature and feeding on juvenile growth ($p = 0.591$; electronic supplementary material, table S3).

Using LSCM, we quantified juvenile tissue thickness and Symbiodiniaceae fluorescence. Feeding increased tissue thickness ($p < 0.001$), but the increase in thickness due to feeding was less under cool temperature (82 µm thicker in fed corals) than under ambient temperature (177 µm thicker in fed corals; $p = 0.037$; figure 3*b*; electronic supplementary material, table S3). Opportunities to feed increased Symbiodiniaceae fluorescence ($p < 0.001$) with a lesser increase in cool temperature (1.6% RI) as compared to ambient temperature (9.0% RI; $p < 0.001$; figure 3*c*; electronic supplementary material, table S3). Tissue thickness and Symbiodiniaceae fluorescence were significantly correlated (Pearson

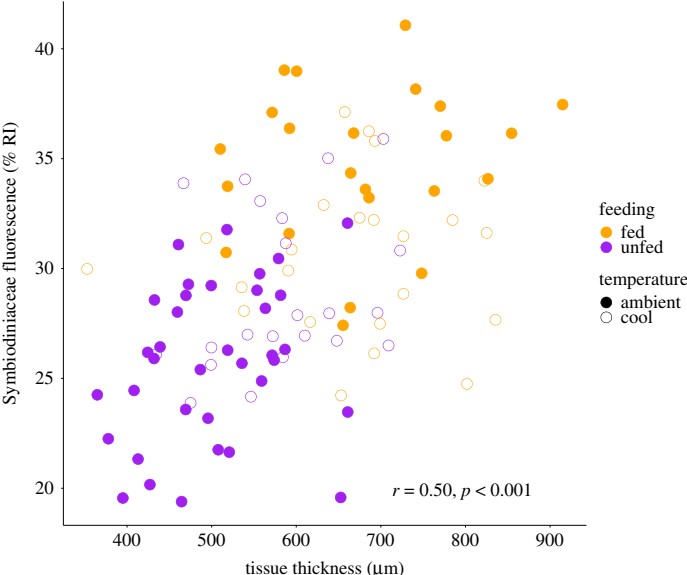

**Figure 4.** Correlation of juvenile tissue thickness (μm) and Symbiodiniaceae fluorescence (colony RI). Feeding treatments indicated by colour (purple = unfed; orange = fed) and temperature indicated by shape (closed points = ambient, open points = cool).

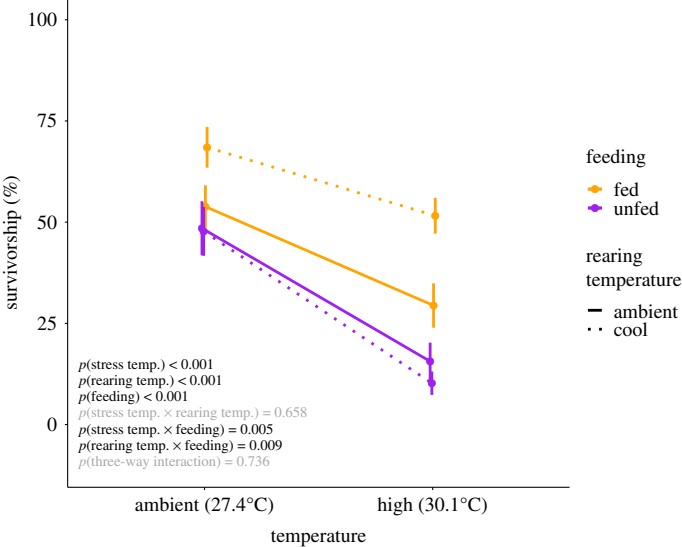

**Figure 5.** Juvenile *Pocillopora acuta* survivorship in ambient and high-temperature treatments during the stress period in feeding treatments (purple = unfed; orange = fed) and from rearing temperature treatments (solid = ambient; dotted = cool). Black text indicates significant effect ($p < 0.05$); grey text indicates non-significant effect ($p > 0.05$).

correlation; $r = 0.50$, $t = 5.83$, d.f. = 102, $p < 0.001$; figure 4). LSCM images of juvenile colonies in each feeding and thermal treatment are shown in figure 3.

## 3.2. Thermal stress period

Juvenile survival was reduced by 28% under high-temperature conditions ($p < 0.001$; figure 5; electronic supplementary material, table S4 and S5). Access to plankton supported juvenile tolerance to thermal stress with 27% higher survival in fed juveniles as compared to unfed juveniles ($p = 0.005$; figure 5; electronic supplementary material, table S5). Fed juvenile corals that were reared in cool temperature prior to thermal stress had the highest survival of any treatment group across temperature treatments ($p = 0.009$; figure 5). Mean survivorship of cool-fed corals was 19% higher in ambient temperature and

33% higher in high temperature as compared to other treatment groups (electronic supplementary material, table S4).

## 4. Discussion

Planktonic prey is an important nutritional source for juvenile *Pocillopora acuta* corals. In this study, fed juveniles demonstrated enhanced performance and thermal tolerance. Importantly, juvenile corals that were fed *and* exposed to a cool thermal regime prior to experiencing thermal stress had the highest survival of all treatment groups. Evidence collected by laser scanning confocal microscopy suggests that tissue thickening combined with regulation of symbiont populations (seen here as a reduction in fluorescence) contributed to observed increases in thermal tolerance in cool-fed juveniles. These results provide new insights on early life-history nutrition in corals and underscore the importance of considering feeding effects in the context of temperature regime. The influence of mixed nutritional modes in supporting the survival of juvenile colonies under ocean warming is critically important to our understanding of shifting recruitment dynamics under climate change.

When fed, juvenile *P. acuta* had enhanced skeletal growth, thicker tissues, higher symbiont fluorescence (figure 3), and were more tolerant to thermal stress (figure 5). Feeding increases calcification and growth, tissue biomass and symbiont carbon fixation in adult colonies [4,12,20] and here we show that juvenile colonies also respond to feeding by increasing tissue thickness and symbiont fluorescence. Enhanced symbiont fluorescence (related to increased cell density and/or photosynthetic pigments [61]) could potentially result in elevated levels of symbiont photosynthesis and translocation of photosynthates [62] in fed juveniles that would increase coral energy storage and serve as a key factor in promoting tissue and skeletal growth [32,58,71]. In addition to the positive effects of feeding on growth and biomass, fed juvenile *P. acuta* demonstrated higher thermal tolerance. By promoting tissue growth and symbiont photosynthesis (reviewed in [4]), feeding facilitates biomass storage and access to energy sources from captured prey that can offset the costs of physiological stress under warming conditions in adult corals [72]. Indeed, access to prey supports the survival of adult corals during bleaching [7,33,37,73,74], and here we show that feeding also provides these benefits in early life history. In particular, tissue thickness was higher in fed juvenile colonies and could contribute to elevated thermal tolerance. Tissue biomass is an energetic store that can be used to meet biological functions and energetic needs—under stress, corals with greater biomass and thicker tissues may gain refuge and protection from oxidative stress and physiological impacts of bleaching [38]. Thicker tissues can also provide tissue shading and protection for symbiont communities during heat stress [38,75,76]. In juvenile *P. acuta* corals with limited energy reserves, even small increases in tissue biomass may provide additional protection during thermal stress events.

In addition to the broad range of physiological benefits of feeding (e.g. growth), the positive impacts of feeding on supporting survival under stress in early life history warrants further attention on optimizing nutritional supplementation in conservation and restoration approaches. The positive effects of plankton feeding on juvenile coral growth, physiology and thermal tolerance support the utilization of nutritional supplementation for juvenile corals as a strategy to maximize growth in early post-settlement and decrease the pressures of size-specific mortality. Increases in juvenile growth in response to feeding have also been demonstrated in *Acropora* spp. and *Pocillopora damicornis,* supporting the utilization of feeding in nursery and restoration settings [56,57] and highlights the importance of monitoring plankton prey communities in natural reef environments and tracking the response of prey communities to increasing stress (e.g. ocean warming). It is therefore critical to monitor and support healthy plankton communities in reef environments to maximize feeding opportunities in the wild (e.g. maintaining and improving water quality) and quantify variation in nutritional profiles of natural (i.e. plankton) and cultured (e.g. *Artemia* nauplii) food sources.

The growth recorded in this study may be higher than would be observed on the reef because benthic competitors were removed during cleaning, and the prey concentrations (greater than 2000 plankters per litre) supplied were at the upper range of natural concentrations. Plankton concentrations in Kāne'ohe Bay of the size class used in our study range from less than 100 [77] to approx. 3000 plankters per liter depending on the season and location of collection [78], and the concentrations used in this study are comparable to those used in previous studies to elicit physiological responses [31,32,79]. It is also important to consider that small juvenile colonies settled in cracks and crevices on reefs [80] may have limited access to prey or access to different prey composition than larger adult colonies with access to a greater area of the water column above the

benthos. Measurement of juvenile feeding rates and carbon utilization from prey sources *in situ* would strengthen our understanding of the role of heterotrophic nutrition on reefs.

Exposure of fed corals to cool water (25.9°C) prior to a thermal stress event increased survival under high temperature. Both in the absence and presence of thermal stress, cool-fed juveniles had higher survival rates than other treatment groups (figure 5). This may be a result of alterations in nutritional dynamics of juvenile corals in this study, which is highlighted by marked increases in tissue thickness and reduction of Symbiodiniaceae fluorescence (figure 3). Specifically, unfed corals retained thicker tissues under 25.9°C as compared to 27.4°C. This could be due to elevated respiratory rate at 27.4°C as compared to 25.9°C, resulting in increased energetic requirements which would capitalize on energy reserves in the host tissue [25,81]. In the presence of an additional food source, tissue biomass and, therefore, the thickness was higher in juveniles that were fed regardless of temperature, due to access to an additional supply of carbon and nutrients available to maintain growth and meet respiratory demand [40].

Access to additional nutrients was also reflected in higher levels of Symbiodiniaceae fluorescence in fed corals, demonstrating cycling of heterotrophically acquired nutritional resources between host and symbiont [20,37]. Reductions in symbiont fluorescence in fed corals exposed to 25.9°C could be due to decreases in host respiratory rate and therefore fewer metabolic byproducts (e.g. inorganic carbon, nitrogen), translocated to symbiont populations from the host, resulting in lower Symbiodiniaceae population fluorescence as compared to 27.4°C juveniles [82]. The elevated survival in cool-fed juveniles suggests that regulation in symbiont fluorescence lowered the risk of mortality under warming conditions. Under high temperature, oxidative stress (e.g. production of reactive oxygen species, ROS) play a role in promoting dysbiosis and causing bleaching [22]. Optimizing nutritional dynamics may therefore provide refuge and protection by limiting the production and impact of ROS with lower symbiont densities, lessening physiological stress and mortality [83]. Cool water exposure can result in the regulation of symbiont communities [84]. When exposed to cold stress, the coral *Stylophora pistillata* demonstrated declines in symbiont densities and chlorophyll pigment concentrations as an acclimation mechanism to limit oxidative stress [84]. The observed reduction in *P. acuta* symbiont fluorescence following cool water exposure may also reflect the regulation of symbiont communities that could limit the buildup of oxidative stress during future exposure to high temperatures and provides new insights into early life history survival strategies.

Similarly, the conservation of energy due to decreased respiratory rates at lower temperatures may reduce the risk of mortality in cool-exposed juveniles [25,81], raising questions on the role of seasonal cool periods in the annual acclimatization of corals [85,86]. As sea surface temperatures continue to rise, temperatures during cooler seasons are becoming more mild in the natural environment [87]. Decreases in the number of cool days reduce seasonal acclimatization windows for corals and, due to the impact of cool thermal exposure on juvenile *P. acuta* in this study, there is a potential for alterations in annual seasonal cycles to impact the structure of juvenile coral communities. Previous work has also raised important questions about the role of cool temperatures in shaping growth and bleaching responses [39,85] and the implications of these effects should be given further attention in future research targeting both early life stages and adult responses. In this study, we can only attribute these observations to a difference in temperature and not as true seasonal effects given that we did not simultaneously manipulate irradiance, water chemistry or other factors that shift by season.

Examining nutritional dynamics and effects of feeding in early life history is critical in both brooding species that reproduce year-round under a range of environmental conditions and in spawning species that reproduce during narrow seasonal windows in order to better understand coral nutritional requirements for survival as the oceans continue to change. Direct measurement of respiration, photosynthesis and translocation of photosynthates in response to feeding in early post-settlement across a range of temperature regimes would provide additional insight into nutritional requirements and thermal tolerance in this life stage. Further, due to the strong effect of feeding on commonly measured responses such as growth, the effects of nutritional supplementation in controlled tank experiments must be carefully considered as coral performance and condition vary significantly in the presence of a food source [88]. Efforts to optimize the propagation of coral recruits for seeding of degraded reef areas are currently underway, and there is an immediate need to improve practices to maximize the survival of juvenile corals in nursery settings [89,90]. High mortality of laboratory-raised juveniles after transplantation to the reef may be ameliorated with nutritional supplementation, and as a result, optimization of food sources provides a method to enhance skeletal and tissue growth in juvenile corals [56,57]. Identifying optimal coordination of nutritional supplementation and thermal conditions may increase the success of juvenile propagation in restoration initiatives by supporting

growth and enhancing resilience to future thermal stress events. As climate change continues to threaten the persistence of coral reefs, understanding early life history survival strategies and identifying conditions that facilitate recruitment is critical to predict reef trajectories in future oceans.

Ethics. All animal collections were approved under the Hawaiʻi Institute of Marine Biology Special Activities Permit no. 2018–03 (Division of Aquatic Resources, Hawaiʻi).

Data accessibility. All data and scripts are publicly available on Zenodo at the following https://doi.org/10.5281/zenodo. 4685948 [70]. The data are provided in electronic supplementary material [91].

Authors' contributions. A.H. conceived and conducted the study, collected data, analysed data, wrote the manuscript and acquired funding. C.J. conducted the study and collected data. A.E. collected data. J.L. and R.G. conceived the study. A.H., C.J., A.E. and J.L. edited and approved the final version of the manuscript.

Competing interests. The authors declare no competing interests.

Funding. This work was supported by the National Science Foundation Graduate Research Fellowship under grant no. DGE1329626; Paul G. Allen Family Foundation; the Philanthropic Education Organization Scholar Award; the University of Hawaiʻi at Mānoa Charles H. and Margaret B. Edmondson Research Fund and the Hawaiʻi Institute of Marine Biology Colonel Willys E. Lord, DVM and Sandina L. Lord Scholarship Fund.

Acknowledgements. We dedicate this manuscript to the life and legacy of our friend and mentor, Dr. Ruth Gates. We thank S. Matsuda, E. Lenz, S. Anderson, J. Davidson, C. Wall, J. Huffmyer and the HIMB facilities staff for assistance in conducting the study. Thank you to the Hawaiʻi Institute of Biology Hagedorn Laboratory for providing the seawater chiller used for this experiment. We also thank the editors and two anonymous reviewers as well as C. Nelson, S. Matsuda, C. Wall, E. Madin, C. Drury and H. Reich for comments that improved the manuscript. This is the Hawaiʻi Institute of Marine Biology contribution #1854 and University of Hawaiʻi at Mānoa School of Ocean and Earth Science and Technology contribution #11334.

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
