## [Peer Review File · Royal Society Open Science]

Review History

Decision letter (RSOS-210644.R0)

Dear Dr Huffmyer

On behalf of the Editors, we are pleased to inform you that your Manuscript RSOS-210644 "Feeding and thermal conditioning enhance coral temperature tolerance in juvenile *Pocillopora acuta*" has been accepted for publication in Royal Society Open Science subject to minor revision in accordance with the referees' reports. Please find the referees' comments along with any feedback from the Editors below my signature.

Please submit your revised manuscript and required files (see below) no later than 7 days from today's (ie 28-Apr-2021) date. Note: the ScholarOne system will 'lock' if submission of the revision is attempted 7 or more days after the deadline. If you do not think you will be able to meet this deadline please contact the editorial office immediately.

on behalf of Dr Cynthia Downs (Associate Editor) and Pete Smith (Subject Editor)
openscience@royalsociety.org

Associate Editor Comments to Author (Dr Cynthia Downs):

Associate Editor

Comments to the Author:

I reviewed this manuscript, the reviewer comments from Proc B, and the response to the reviewer comments. The authors have made sufficiently address the reviewers' comments and I enjoyed reading this well-written manuscript of a well-designed study. The changes broaden the implication of the research by putting it into a broader framework and support the results since they are robust to different statistical frameworks. I have no major edit and only one minor comment. Change the colors in Fig 5 to match those in Fig 3 & 4.

===PREPARING YOUR MANUSCRIPT===

While not essential, it will speed up the preparation of your manuscript proof if you format your references/bibliography in Vancouver style (please see

<https://royalsociety.org/journals/authors/author-guidelines/#formatting>). You should include DOIs for as many of the references as possible.

===PREPARING YOUR REVISION IN SCHOLARONE===

<https://royalsociety.org/journals/authors/author-guidelines/#data>. You should ensure that you cite the dataset in your reference list. If you have deposited data etc in the Dryad repository,

please only include the 'For publication' link at this stage. You should remove the 'For review' link.

Author's Response to Decision Letter for (RSOS-210644.R0)

See Appendix A.

Decision letter (RSOS-210644.R1)

Dear Dr Huffmyer,

It is a pleasure to accept your manuscript entitled "Feeding and thermal conditioning enhance coral temperature tolerance in juvenile *Pocillopora acuta*" in its current form for publication in Royal Society Open Science. The comments of the reviewer(s) who reviewed your manuscript are included at the foot of this letter.

on behalf of Dr Cynthia Downs (Associate Editor) and Pete Smith (Subject Editor)
openscience@royalsociety.org

Appendix A

Associate Editor Comments to Author (Dr Cynthia Downs):

Associate Editor

Comments to the Author:

I reviewed this manuscript, the reviewer comments from Proc B, and the response to the reviewer comments. The authors have made sufficiently address the reviewers' comments and I enjoyed reading this well-written manuscript of a well-designed study. The changes broaden the implication of the research by putting it into a broader framework and support the results since they are robust to different statistical frameworks. I have no major edit and only one minor comment. Change the colors in Fig 5 to match those in Fig 3 & 4.

We thank the associate editor for their positive review of our manuscript and thank the previous reviewers for comments that have improved our manuscript. In this revision, we have changed the colors of Fig 5 to match those in Fig 3 and 4 by using color to denote feeding treatment (orange = fed, purple = unfed) and using line type to denote temperature treatment (solid = ambient, dotted = cool). We have revised the figure captions to reflect this change in color. We hope that our manuscript is now suitable for publication in Royal Society Open Science.